# MCML-BF: A Metal-Column Embedded Microstrip Line Transmission Structure with Bias Feeders for Beam-Scanning Leakage Antenna Design

**DOI:** 10.3390/s24113467

**Published:** 2024-05-28

**Authors:** Shunhu Hou, Shengliang Fang, Youchen Fan, Yuhai Li, Zhao Ma, Jinming Li

**Affiliations:** 1Graduate School, Space Engineering University, Beijing 101416, China; hshhsc2022@163.com (S.H.); lyth17739854226@163.com (Y.L.); nuaamazhao@163.com (Z.M.); 2School of Space Information, Space Engineering University, Beijing 101416, China; love193777@sina.com (Y.F.); 13611214081@163.com (J.L.)

**Keywords:** beam scanning, periodic leakage antenna, holographic principle, liquid crystal, bias feeder

## Abstract

This article proposes a novel fixed-frequency beam scanning leakage antenna based on a liquid crystal metamaterial (LCM) and adopting a metal column embedded microstrip line (MCML) transmission structure. Based on the microstrip line (ML) transmission structure, it was observed that by adding two rows of metal columns in the dielectric substrate, electromagnetic waves can be more effectively transmitted to reduce dissipation, and attenuation loss can be lowered to improve energy radiation efficiency. This antenna couples TEM mode electromagnetic waves into free space by periodically arranging 72 complementary split ring resonators (CSRRs). The LC layer is encapsulated in the transmission medium between the ML and the metal grounding plate. The simulation results show that the antenna can achieve a 106° continuous beam turning from reverse −52° to forward 54° at a frequency of 38 GHz with the holographic principle. In practical applications, beam scanning is achieved by applying a DC bias voltage to the LC layer to adjust the LC dielectric constant. We designed a sector-blocking bias feeder structure to minimize the impact of RF signals on the DC source and avoid the effect of DC bias on antenna radiation. Further comparative experiments revealed that the bias feeder can significantly diminish the influence between the two sources, thereby reducing the impact of bias voltage introduced by LC layer feeding on antenna performance. Compared with existing approaches, the antenna array simultaneously combines the advantages of high frequency band, high gain, wide beam scanning range, and low loss.

## 1. Introduction

Antennas are vital in wireless communication, transmitting and receiving electromagnetic waves [1,2,3,4,5,6]. Different applications require various antenna functionalities, among which beam scanning performance is crucial to the system function [7,8,9,10]. Since the 1940s, the leaky-wave antenna has garnered significant attention due to its exceptional radiation characteristics and impressive beam scanning capability [11,12,13]. Recently, various leaky-wave antennas have emerged, offering advantages such as high efficiency, low profile, compact size, and simplified feed structure [14,15]. Nevertheless, conventional leaky-wave antennas are limited to frequency beam scanning, which restricts their usability in scenarios with limited spectrum resources. Therefore, there is a need to explore fixed-frequency beam scanning leakage antennas [16,17,18,19,20].

Two methods are commonly used in the production of fixed-frequency beam-scanning leakage antennas, i.e., loading active devices and utilizing tunable materials. The active device loading method mainly uses PIN diodes and varactor diodes [21,22,23,24]. The diode based fixed-frequency scanning leakage antenna has the advantages of stable electromagnetic performance and low cost, but its main disadvantage is that it is not used in high frequency bands, and it cannot be used in millimeter bands or even higher frequency bands. Tunable materials include ferrite, graphene, and LC materials. Nil Apaydin and other scholars proposed a fixed-frequency scanning leakage antenna based on ferrite [25], which achieves beam scanning by applying a static magnetic field at 1.79 GHz. The scanning angle can reach 80°. However, this antenna has a large volume, high power consumption, and a complex external magnetic field structure, which is not conducive to system integration. Esquius Morote proposed a sine modulated graphene leakage antenna with fixed-frequency beam-scanning capability. The antenna operates at terahertz frequencies and sinusoidally modulates the surface reactance of graphene through the field effect of graphene, providing multifunctional beam scanning capability [26]. However, most graphene-based antennas are currently in the theoretical stage, and their electrical modulation characteristics can only be used in terahertz frequency bands [27,28,29]. LC materials are gaining popularity, as they make fixed-frequency beam scanning possible when used in conjunction with leaky-wave antennas [30,31,32]. This is due to the fact that, compared with other electronically controlled materials, LC materials can be utilized from Ku-band all the way up to the optical band, and the insertion loss decreases as the frequency band rises. Therefore, LC materials have shown promise in the design of high-frequency microwave devices [25,33,34]. Significant progress has been made in single-beam scanning leaky-wave antennas using LC materials [35,36,37,38]. In the initial stage, the gain of beam-scanning leakage antennas based on LC materials is primarily low. Yan Gao proposed an electron beam scanning leakage antenna based on composite left- and right-handed rectangular waveguides, operating at 9.7 GHz and offering a beam scanning range of −19 ° to 12° [39]. Yaling Liu designed an LC-based leakage antenna featuring 13 rectangular slits, operating at 10.4 GHz, with a beam pointing range from 30° to 60° and a gain of up to 5 dB [40]. Feng Gao developed a microstrip line cell with LCs, operating at 19.8 GHz and 21.8 GHz. This design yielded a holographic metamaterial antenna with reconfigurable directional maps thanks to the periodic arrangement of 16 cells. The antenna offered a scanning range of −25° to 60° and a gain of up to 8 dB [41].

Subsequent researchers have focused on beam gain and achieved high-gain beam scanning. The authors of [42] used a half-mode comb substrate integrated waveguide circuit as a leaky transmission line, achieving beam scanning from 2° to 20° at 21.5 GHz with a peak gain of up to 12 dB. While their technology has a high gain, the beam scanning range remains narrow, and the operating frequency band is limited. With further development, the operating frequency of leakage antennas is progressing toward higher frequency bands, resulting in numerous notable research achievements. Qi Liu designed an LC-based beam-scanning ML leakage antenna comprising 64 units operating at 30 GHz. This design achieved beam scanning from −27° to 38°, with a gain ranging from 9.5 dB to 12.5 dB [43]. However, the scanning range of the leakage antenna was narrow. Weiyi Zhang proposed an LC substrate integrated waveguide leakage antenna based on holographic theory, operating at 35 GHz. This design achieved a beam scanning range from −45° to 51°, with a maximum gain exceeding 9 dBi and a reflection coefficient below −10 dB [44]. A microstrip line leakage antenna based on the etched complementary open resonant ring of LC materials is proposed in [45]. The antenna works in the Ka-band, and by arranging 56 antenna units to form a periodic leakage antenna, it can achieve a scanning angle between −53° and 60° at 34.7 GHz, and the gain can reach 12.63 dB.

Current research on beam scanning leakage antennas based on LC materials has achieved an excellent level; these advances have the advantages of wide beam scanning range, high gain, and high-frequency band. The authors of [45] achieved excellent beam scanning performance. Low attenuation loss transmission of electromagnetic waves was accomplished by embedding two rows of metal columns in the medium on the basis of a traditional ML transmission structure. However, that study did not consider the bias feeder problem introduced by the LC layer feed, only illustrating that the bias feed has a negligible effect; as such, those authors did not further investigate how much the antenna radiation performance was affected by the LC layer feed voltage. In this article, we follow the transmission structure proposed in this thesis. To realize the feeding of the LC layer more conveniently, we encapsulated the LC layer in the transmission medium of the microstrip line and the metal ground plane. Introducing a bias feeder on the ML [43,46] forms a voltage difference with the metal grounding plate, thus allowing voltage control of the LC layer. To further minimize the impact of the introduced bias voltage on the RF source and the radiation performance of the antenna, we adopted a sector-blocking bias feed structure [47,48]. This structure can significantly reduce the influence between the two sources, thus reducing the impact of the bias voltage introduced due to the LC feed on the antenna performance.

## 2. Liquid Crystal and Holographic Antenna Principle

### 2.1. Liquid Crystal

LC is a kind of organic compound between a crystal and a liquid, with both the fluidity of liquid and the anisotropy of crystal [49]. When DC voltage or low-frequency AC voltage is applied to both ends of the LC layer, the long-axis direction of the LC molecules will change, as will the dielectric constant of the LC [50]. When the electromagnetic wave passes through the LC layer as the transmission medium of a microwave device, the transmission characteristics of the electromagnetic wave will change with the change of voltage, thereby changing the transmission characteristics of the electromagnetic wave.

There are many ways to categorize LC materials. According to the difference in how the molecules are arranged within the LC, LC materials can be classified into three types: nematic LCs, near-crystalline LCs, and cholesteric LCs. Nematic LCs are composed of polar rod-shaped molecules with a dipole moment. The center of gravity of each LC molecule has no rules, so the mutual binding between molecules is relatively small, the viscosity is low, and it is easy to rotate under external force [51]. In the design of microwave millimeter waves, nematic LCs are usually selected. The LC material chosen in the design of this paper was also a nematic LC. The electromagnetic properties (behavior) of nematic LCs are modelled as vectors [52,53,54]:(1)$$\hat{\varepsilon }=\varepsilon_{⊥}^{ℜ,ℑ}\bar{I}+(\varepsilon_{\left|\right|}^{ℜ,ℑ}-\varepsilon_{⊥}^{ℜ,ℑ})\vec{e}\times {\vec{e}}^{T}$$
where the unit vector $\vec{e}$, called the director, represents the average orientation of LC molecules, ${\vec{e}}^{T}$ is the transposition of $\vec{e}$, $\hat{\varepsilon }$ is the LC dielectric permittivity tensor, $\bar{I}$ is the identity matrix, and $\varepsilon_{⊥}^{ℜ,ℑ}$ ($\varepsilon_{\left|\right|}^{ℜ,ℑ}$) is the LC complex permittivity felt by the perpendicular (parallel) electric field component to the director. The complex permittivities are:(2)$$\varepsilon_{⊥,||}^{ℜ,ℑ}=\varepsilon_{⊥,||}(1-j\tan\delta_{⊥,||})$$
where $\tan\delta_{⊥,||}$ are the LC loss tangents. The tuning ability of the microwave device depends not only on its design but also mainly on the degree of the tuning ability of the tunable material. Generally, the absolute and relative tuning abilities of LC material are defined as:(3)$$\Delta \varepsilon =\varepsilon_{\left|\right|}-\varepsilon_{⊥}$$
(4)$$\tau =\frac{\Delta \varepsilon }{\varepsilon_{\left|\right|}}$$

Table 1 shows the parameters of different nematic LC materials at room temperature (20 °C). This paper adopted the No. GT3-23001 LC material with a more extensive tuning range. The relative dielectric constant of the LC material can be changed from 2.5 to 3.3 under applied voltage.

### 2.2. Holographic Antenna Principle

Beam scanning may be realized based on the holographic antenna principle [55]. Holographic technology was first used in optics; it is a kind of light that records all the information about an object and can restore an image of the object’s technology. Holographic technology mainly includes shooting and imaging, i.e., two processes. Shooting uses reference light and object light interference in the holographic film to record all the information. Imaging refers to the use of reference light to irradiate the holographic film process. Due to the diffraction phenomenon of light, the reference light can be restored to the original image of the object through a holographic film. The primary focus of holographic antenna design is to generate a holographic image that records the information of the target wave. The target wave can be restored by illuminating the holographic image with a reference wave source [56,57]. The reference wave equation, target wave equation, and interference wave equation for energy recording can respectively be expressed as follows:(5)$$\psi_{ref}\left(y_{n}\right)=\exp\left(-j\tilde{k}y_{n}\right)$$
(6)$$\psi_{obj}\left(y_{n},\theta_{0}\right)=\exp\left(-jk^{\prime }\sin(\theta_{0})y_{n}\right)$$
(7)$$\psi_{inf}\left(y_{n},\theta_{0}\right)=\psi_{ref}^{*}\left(y_{n}\right)\psi_{obj}\left(y_{n},\theta_{0}\right)=\exp(j(\tilde{k}-k^{\prime }\sin(\theta_{0}))y_{n})$$
where $y_{n}$ represents the position information of the recording point on the holographic structure, $\tilde{k}$ represents the propagation constant of the reference wave, and $k^{\prime }$ represents the propagation constant of the object wave, which is the free-space propagation constant. $\theta_{0}$ represents the beam direction of the object wave, which is the expected beam direction.

The holographic antenna principle is a technique that achieves the desired beam pointing by combining the amplitude-weighting technique, which assigns a weighted value to each antenna element based on its contribution to the expected beam direction. By varying the radiated energy of each antenna element, the cells that contribute to the expected beam direction are given a higher radiated power, while those that do not contribute are given a lower radiated power. This may be achieved by taking the real part of the Equation (3) to get $\cos[(\tilde{k}x_{n}-k^{\prime }\sin(\theta_{0})y_{n})]$ and performing binary discretization processing. In the amplitude-weighted technique, a threshold is often used to determine which parts of the antenna signal should be retained and which should be suppressed. The choice of this threshold can affect the effectiveness of signal processing, so it needs to be chosen carefully; 0.5 is usually chosen, because this value balances the retention and rejection of the antenna signal. If the threshold is too low, too much noise and clutter will be retained, resulting in inaccurate signal processing results. If the threshold is too high, too many valuable signals will be suppressed, resulting in severe signal attenuation and ineffective subsequent processing. Therefore, if $\left[\cos[(\tilde{k}x_{n}-k^{\prime }\sin(\theta_{0})y_{n})]+1\right]/2<0.5$, the state of the cell at that position is “closed”. It is considered that this unit is not conducive to “reproducing” the antenna beam direction, and the excitation amplitude value of this unit is set to “0”. If $\left[\cos[(\tilde{k}y_{n}-k^{\prime }\sin(\theta_{0})y_{n})]+1\right]/2>0.5$, the state of the cell at the position is “open”. It is considered that this unit is conducive to “reproducing” the antenna beam direction, and the excitation amplitude value of this unit is set to “1”.

## 3. Design and Simulation Results

### 3.1. Antenna Unit Cell

The leakage antenna designed in this paper utilizes an ML transmission structure. The ML transmission structure has the advantages of low cost, easy integration, and high flexibility, but the attenuation loss (such as dielectric loss, conductor loss, and scattering loss) is large. Recently, the substrate-integrated waveguide (SIW) transmission structure has been popularized for low energy loss. It connects the upper and lower layers of metal by placing two rows of metal columns in the substrate to realize electromagnetic wave transmission [8,58,59]. However, this structure has the disadvantages of a complicated fabrication process and poor flexibility in unit configuration. To combine the advantages of the two structures, we propose a metal-column embedded microstrip line (MCML) transmission structure based on the traditional ML structure by borrowing the structural characteristics, i.e., by placing two rows of metal posts in the SIW [45]. It is worth noting that, unlike SIW, the metal columns of the MCML structure do not play the role of connecting the upper and lower metal layers. Rather, they have two main roles: (1) limiting the electromagnetic wave propagation region, so that the electromagnetic wave is concentrated in the radiation region, thereby transmitting the electromagnetic wave more efficiently and decreasing the dissipation of energy, thus reducing the radiation loss; and (2) reducing the attenuation loss and improving the efficiency of energy radiation.

Figure 1 depicts the leakage antenna unit comprising a copper ground floor, a dielectric substrate, an ML etched with the CSRR structure, and an LC layer. The dielectric substrate consists of a 1.5-mm-thick glass with a relative dielectric constant of 5.5. A 0.035-mm-thick copper plate is placed at the bottom of the dielectric substrate as the ground floor. The ML etched the CSRR is made of metallic copper with a thickness of 2 μm and is topped with a 0.2 mm-thick glass layer. The LC layer is encapsulated in a dielectric substrate and filled between the ML and the ground metal plate. By introducing a bias voltage to form a pressure difference between the ML and the metal plate, the direction of the LC layer molecules can be controlled to change the antenna resonant frequency. Two rows of metal columns are placed on a dielectric substrate for a more efficient transmission of electromagnetic waves. The detailed dimensions of the antenna structure are shown in Table 2.

The Complementary Split Ring Resonator (CSRR) is a complementary form of the Split Ring Resonator (SRR) [60]. The SRR comprises two metal rings with identical centers and back-to-back openings and has negative magnetic permeability characteristics similar to those observed in CSRR. When excited by an axial magnetic field, the CSRR can be regarded as a magnetic dipole. At the same resonant frequency, the CSRR composed of multiple metal rings with the same center of the circle is compact compared to that composed of a single metal ring, rendering the device more miniaturized [61]. The electric and magnetic fields at the CSRR-shaped gap on the microstrip line are shown in Figure 2. The E field intensity is greatest at the open resonant ring of the CSRR slot (Figure 2a). This time-varying e-field can be expressed as a capacitance by the extended Ampere law. At the same time, the magnetic field is mainly distributed in the upper and lower sides of the outer ring of the gap (Figure 2b), which can be expressed as inductance by Faraday’s induction law. Therefore, CSRR gaps can be modelled with inductance (L), radiation resistance (R), and capacitance (C). By changing the geometrical parameters of the gap, L, R, and C will change, and the resonant frequency and coupling strength will also change.

The RF energy input to the antenna unit is set to *P_in_*, the energy radiated from the CSRR slit to *P_rad_*, the dielectric loss to *P_die_*, the scattering loss to *P_sca_*, and the conductor loss to *P_con_*. *P_die_*, *P_sca_*, and *P_con_* are collectively referred to as the unwanted attenuation loss, denoted by *P_att_*. *S*_11_ represents the reflection coefficient of the antenna, and *S*_21_ denotes the transmission coefficient. The radiation efficiency of the antenna can be expressed as:(8)$$\eta_{rad}=\frac{P_{loss}}{P_{in}}=\frac{P_{rad}+P_{sca}+P_{con}+P_{die}}{P_{in}}=\frac{P_{rad}+P_{att}}{P_{in}}=1-{\left|S_{11}\right|}^{2}-{\left|S_{21}\right|}^{2}$$

To verify the advantages of the MCML transmission structure, we removed the CSRR slot of the ML in the antenna structure to avoid the energy leakage from the microstrip line to free space, i.e., *P_rad_* was approximated to be 0. At the same time, the antenna element of the ML structure was used as a contrast structure. The difference between the two structures was that the MCML structure had two rows of metal columns. We used HFSS simulation software (Version 2020R1) to simulate and obtain the *S*_11_ (Figure 3a) and *S*_21_ (Figure 3b) comparison diagrams of the two structures. According to Figure 3a, there was very little difference in the *S*_11_ parameters between the two structures. At 38 GHz, the reflection coefficient of the MCML structure was smaller, indicating that the impedance-matching performance of the MCML structure was better at this frequency. From Figure 3b, it can be concluded that the change of LC dielectric constant had little effect on the transmission coefficient (*S*_21_), and the *S*_21_ parameters of the MCML structure was much higher than that of the ML structure in the whole frequency range of 35–45 GHz. According to Equation (8), *P_rad_* is approximated to be 0. The *P_in_* of two structures was the same, the difference of *S*_11_ was negative, and the S_21_ parameters of the MCML were much higher than those of the ML, which indicated that the attenuation loss *P_att_* of the ML structure was larger. Meanwhile, the MCML structure could reduce the useless attenuation of electromagnetic waves and improve energy utilization efficiency.

The interaction between the metal columns and the transmission medium in MCML structures can increase the energy coupling effect. By reasonably designing the shape and position of the metal columns, the propagation path of electromagnetic waves can be controlled, allowing energy to be more concentrated and transmitted to the target area, reducing beam diffusion. Figure 4a–d represent the electric field energy maps of ML and MCML at the output wave port, as well as the electric field vector maps of the antenna surface, respectively. Figure 4b shows that the MCML structure can concentrate electromagnetic wave energy more in the middle radiation region, reduce the beam diffusion degree, and increase the electric field energy value. Figure 4d shows more intuitively that the MCML structure can better control the transmission of electromagnetic waves, resulting in higher peak values of the electric field vector.

A simulation of the MCML structure yielded the *S*_11_ parameters (Figure 5a) and *S*_21_ parameters (Figure 5b). The antenna design employed GT3-23001 LC material. Altering the voltage applied to the upper and lower surfaces of the LC material induces a change in its dielectric constant from 2.5 to 3.3. This variation in dielectric constant leads to a shift in the resonant frequency of the resonator, subsequently impacting the energy coupled out from the slots and the radiation efficiency.

When the dielectric constant of the LC is 2.5, the antenna exhibits significant resonance characteristics at 38 GHz. The *S*_11_ parameters corresponding to this frequency are below −20 dB, indicating that the antenna unit has excellent impedance-matching performance. Figure 6 shows the dispersion curves of the antenna unit in both modes. The leaky-wave antenna designed in this paper is arranged periodically along the Y-axis direction, so the master–slave boundary condition is set in the Y direction. The angle parameter is taken from 0° to 180° in steps of 10°, and the dispersion curve is the result of a line scanning from 0°–180° parameter, with the vertical axis being the frequency. The three-dimensional far-field radiation diagram of the antenna is shown in Figure 7a. Due to the limited radiation energy of a single antenna unit, the antenna gain is low. Figure 7b shows the E-plane radiation map at different azimuth angles (*phi*), with slight changes in the E-plane radiation map as the azimuth angle changes.

### 3.2. Antenna Array

The design and optimization of a leakage antenna cell with an MCML structure was described in the previous Section 3.1. This device may be used for 1D array configuration (see Figure 7). The radiation emitted by this array into free space can be conceptualized as a periodic LWA.

The structure of the periodic leakage antenna undergoes periodic changes as a whole. Considering an infinitely long periodic leakage antenna, the electromagnetic waves propagate in the *y* direction. The period length of the antenna is denoted as *d* and the propagation constant as $\beta_{y}$, and the attenuation constant is assumed to be 0, with only a phase difference, described as $\exp(\beta_{y}yd)$. If the electric field in the first period is represented by *E*(*z*), then the electric field at that same position in the subsequent period is represented by $E(z)\exp(\beta_{y}yd)$. Formula (9) represents the field distribution in the leakage structure.
(9)$$E\left(x,y,z\right)=E_{d}(x,y,z)e^{-j\beta_{y}y}$$
where *E_d_*(*x*,*y*,*z*) is a function of period length *d*, which can be expanded by Fourier series as:(10)$$E_{d}(x,y,z)=\sum_{-\infty }^{\infty }E_{dn}(x,z)e^{-j\frac{2n\pi }{d}y}$$

Substituting Formula (10) into Formula (9), the expression for the electric field distribution is given in Formula (11).
(11)$$E(x,y,z)=\sum_{-\infty }^{\infty }E_{dn}(x,z)e^{-j\beta_{n}y}$$

In the above formula,
(12)$$\beta_{n}=\beta_{y}+2n\pi /d$$

It can be observed from Formulas (11) and (12) that the electric field *E*(*x*,*y*,*z*) is comprised of multiple harmonics, each with corresponding propagation constant. These propagation constants are determined by the fundamental mode propagation constants and the period between elements.

In accordance with the radiation conditions of leakage antennas, for periodic leakage antennas to radiate in the region where *z* is greater than 0, wave number *k_zn_* must also be greater than 0. Otherwise, electromagnetic waves will rapidly decay along the *z* direction. The free space propagation constant is denoted as *k*_0_, and the relationship between wave number *k_zn_* and phase constant *β_n_* are obtained as shown in Formula (13).
(13)$$k_{zn}^{2}=k_{0}^{2}-\beta_{n}^{2}=k_{0}^{2}-{\left(\beta_{y}+\frac{2n\pi }{d}\right)}^{2}$$

The radiation condition under which harmonics can be obtained is:(14)$$-k_{0}<\beta_{y}+\frac{2n\pi }{d}<k_{0}$$

The fundamental mode of a periodic leakage antenna is typically a slow wave, with the propagation constant of the fundamental mode being *β_y_* > *k*_0_. The radiation condition for the *n* harmonics can be derived from Formula (14).
(15)$$-\frac{2n\pi }{\beta_{y}+k_{0}}<d<-\frac{2n\pi }{\beta_{y}-k_{0}}$$

Formula (15) indicates that *d* becomes negative when *n* is greater than 0, which is not practical. When *n* is less than 0, the antenna can generate radiation for *n* harmonics by appropriately designing the period *d*. In this case, the value of *β_n_* can be greater than 0, equal to 0, or less than 0. Therefore, the periodic leakage antenna exhibits zero-cross-scanning characteristics.

Arranging the 72 antenna units in one dimension, according to the principle of periodic leakage antennas, the choice of unit spacing greatly influences the antenna’s radiation characteristics. Take, for example, *n* = −1, so that the antenna array can generate −1 harmonic radiation. After optimizing the design, the spacing between units is 2 mm, and the antenna length is 123 mm, as shown in Figure 8.

We set the operating frequency to 38 GHz and simulated the antenna performance using HFSS. The preset angles for beam scanning were −55°, −30°, −20°, 0°, 20°, 30°, and 60°, respectively. Table 3 displays the antenna unit‘s encoding sequence “0/1”, corresponding to each beam direction. The relative dielectric constant of the LC corresponding to “0” was 3.3, and the relative dielectric constant of the LC corresponding to “1” was 2.5. Figure 9 illustrates the radiation spatial beam gain maps corresponding to different beam directions.

Figure 10 shows that the reflection coefficient (i.e., the *S*_11_ parameters) was essentially below −10 dB in the 36 GHz–40 GHz frequency range over a beam sweep from −52° to 54°, indicating an excellent impedance match.

As shown in Figure 9, the different beam directions aligned with expectations and exhibited a gain above 10 dB overall. Particularly at an angle of 30°, there was an angle deviation of just 0.4°, and a gain reaching as high as 11.65 dB could be achieved. Simulation experiments showed that the angle scanning range of the antenna could not reach −55° and 60°, and the angle scanning between −52° and 54° could be realized when working at 38 GHz. Table 4 shows specific angle deviations and main beam gains for different directions.

### 3.3. Bias Feeder Design and Comparative Experiments

During the antenna testing, it was necessary to apply appropriate bias voltages between the metal ground plate and the ML to control the dielectric constant of the LC at each unit. This required loading a suitable bias feeder onto the antenna. However, introducing bias feeders to the antenna unavoidably affects its radiation characteristics to some extent. Hence, this section primarily investigates how to load the bias feeder onto the antenna structure and the actual impact on the antenna after loading the bias feeder.

The design of bias feeders mainly involves two aspects: the position of the loading of the bias feeder and the structure of the bias feeder. Firstly, it is necessary to determine the position of the bias feeder by applying bias voltage on the ML to control the variation of the LC dielectric constant. Since electromagnetic waves primarily radiate into free space through the CSRR slot on the ML, determining the position of the bias feeder on the ML is particularly important. As shown in Figure 2, the radiated energy mainly distributes around the open resonant rings of the CSRR structure. Therefore, it is not advisable to use the long side of the ML as the feeding point for the bias feeder, as this would significantly impact the radiation characteristics. Thus, the bias feeding point was placed on the short side of the ML, as illustrated in Figure 11. The width of the bias feeder was set to *w_bias_*, and the distance from the long side of the rectangular patch was set to *d_bias_*. We next determined the position that minimally affected the antenna’s radiation characteristics by simulating and analyzing different lengths of *d_bias_* values and *w_bias_* values.

We first set *d_bias_* at 1 mm and set *w_bias_* values at 0.05 mm, 0.055 mm, 0.06 mm, 0.065 mm, 0.07 mm, 0.075 mm, 0.08 mm, 0.085 mm, 0.09 mm, 0.095 mm, and 0.1 mm, respectively, for simulation of the antenna unit. By scanning the values of *w_bias_*, the minimum S_31_ curve was identified in the simulation results. A smaller S_31_ value indicates a more minor influence between the bias feed voltage and the RF source, resulting in a lesser impact on the antenna radiation characteristics introduced by the bias feeder. Figure 12 demonstrates that as the width of the bias feeder decreased, the *S*_31_ parameters decreased, indicating a lesser influence between the bias feed voltage and the RF source. However, considering practicality, excessively narrow *w_bias_* values would pose challenges in processing and storage. Therefore, a value of 0.07 mm was applied.

We next set *w_bias_* at 0.07 mm and *d_bias_* values at 0 mm, 0.1 mm, 0.2 mm, 0.3 mm, 0.4 mm, 0.5 mm, 0.6 mm, 0.7 mm, 0.8 mm, 0.9 mm, 1 mm, 1.1 mm, 1.2 mm, 1.3 mm, 1.4 mm, 1.5 mm, 1.6 mm, 1.7 mm, 1.8 mm, and 1.9 mm, respectively, for our simulation of the antenna unit. By scanning the values of *d_bias_*, the minimum S_31_ parameter value was found in the simulation results. As shown in Figure 13, when *d_bias_* was set to 1.5 mm, the S_31_ parameter value was minimized at 38 GHz.

To further minimize the impact of RF signals on the DC source while avoiding the DC bias voltage becoming part of the RF circuit as much as possible, ensuring the stable operation of the antenna, it was necessary to place a blocking structure at the appropriate position of the bias feeder. Assuming the bias feeder as a transmission line, for RF signals, the end connecting to the ML served as the RF input port. According to transmission line principles, when the electrical length of the ML is λ_g_/4 (where λ_g_ is the guided wavelength), the output end becomes short-circuited, and the input end exhibits a high impedance state. By placing a low impedance component at λ_g_/4, it can be approximately considered that the transmission line is short-circuited at λ_g_/4, blocking the RF signal from entering the bias feeder. We utilized a sector-shaped structure to achieve this blocking function. This sector-shaped patch can be equivalent to a low-impedance component. As shown in Figure 14, we placed the sector-shaped patch at a distance of λ_g_/4 from the narrow edge of the ML, i.e., *d_sec_
*= λ_g_/4, with the radius *r_sec_* and an angle α of 90°.

We then set *d_bias_* at 1.5 mm and determined the values of *d_sec_* and *r_sec_*. Due to the loading of the bias feeder, the actual value of the guided wavelength of the antenna unit could not be precisely calculated. The value of *d_sec_* had to be scanned to obtain the minimum S_31_ curve in the simulation results. Before scanning the value of *d_sec_*, it was necessary to determine the scanning range. Calculating the guided wavelength at 38 GHz without loading the bias feeder yielded a value of 3.6 mm. Therefore, the range of *d_sec_* was set from 0.5 mm to 1.5 mm, with a scanning interval of 0.1 mm. Initially, *r_sec_* was set to 0.25 mm. The scanning results are shown in Figure 15a. When the value of *d_sec_* was 1.4 mm, the S_31_ parameter was minimized at 38 GHz. With *d_sec_* fixed at 1.4 mm, we determined the value of *r_sec_*. The range of *r_sec_* was set from 0.1mm to 0.35 mm. The scanning results, as shown in Figure 15b, indicated that the *S*_31_ parameter was minimized at 38 GHz when the value of *r_sec_* was 0.23 mm.

Finally, we set *w_bias_* to 0.07 mm and *d_bias_* to 1.5 mm, *d_sec_* to 1.4 mm, and *r_sec_* to 0.23 mm. After determining the position and structure of the bias feeder, the antenna cell with the bias feeder was assembled into an array, and the overall structure of the antenna was simulated to analyze the impact of the bias network on antenna radiation. Preset angles were set to −55°, −30°, 0°, 30°, and 60°. We employed the binary beam control method to reconstruct the target beam. Figure 16 and Figure 17 show the beam scanning results and reflection coefficients, respectively.

Figure 16 indicates that the angle scanning range of the antenna array remained within the range of −52° to 54° after adding the bias feeder network. However, there was a slight deviation in the specific angle scanning range, from −51.7° to 53.8°. The impedance-matching performance was satisfactory. Table 5 provides particular angle deviations and main beam gains for different beam directions.

Table 5 provides the specific beam scanning range and each angle’s gain. Compared to the antenna gain values presented in Table 4, there was a certain degree of decrease in the actual gain at each angle, with the maximum decrease being 0.758 dB and an average reduction of 0.451 dB. Overall, the impact on the antenna radiation performance after adding bias feeders was minimal, which is crucial for practical engineering applications.

## 4. Conclusions

This paper presents a fixed-frequency beam scanning leakage antenna based on LC. The antenna used 72 CSRRs arranged periodically to achieve beam scanning. The LC layer was encapsulated in the transmission medium between the ML and the metal ground floor. By adding a bias voltage to the ML, the voltage difference between the upper and lower surfaces of the LC layer could be formed to realize the feeding of the LC layer. Combined with the holographic principle, the antenna could achieve continuous beam scanning at 38 GHz from −52° to 54°. We designed the sector-blocking bias feeder structure to minimize the interaction between the introduced bias voltage and the RF source. The beam scanning performance of the antenna was re-simulated after the addition of the bias network. Experiments showed that the bias feeder’s effect on the antenna radiation performance was tiny.

## Figures and Tables

**Figure 1 sensors-24-03467-f001:**
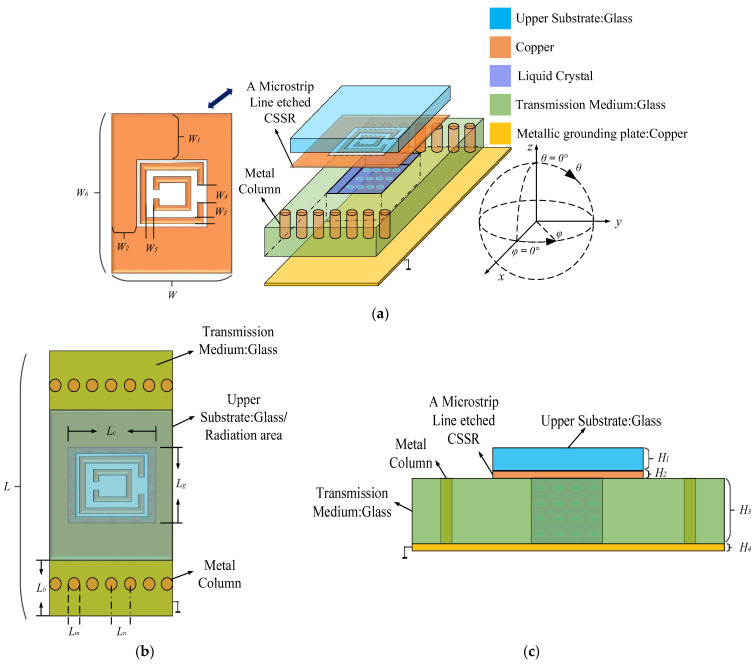
The antenna cell structure. (**a**) Main view; (**b**) Top view; (**c**) left view.

**Figure 2 sensors-24-03467-f002:**
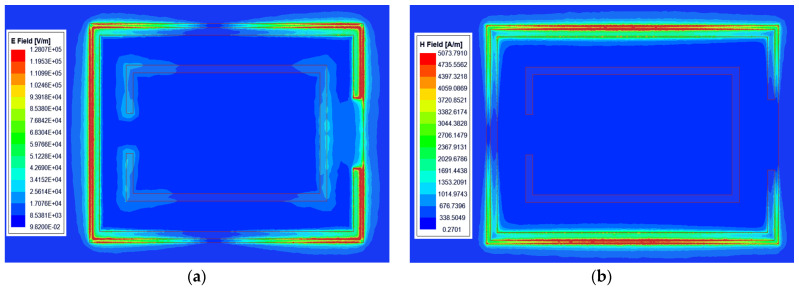
Electric and magnetic field energy distribution at the CSRR slot. (**a**) Electric field energy distribution; (**b**) Magnetic field energy distribution.

**Figure 3 sensors-24-03467-f003:**
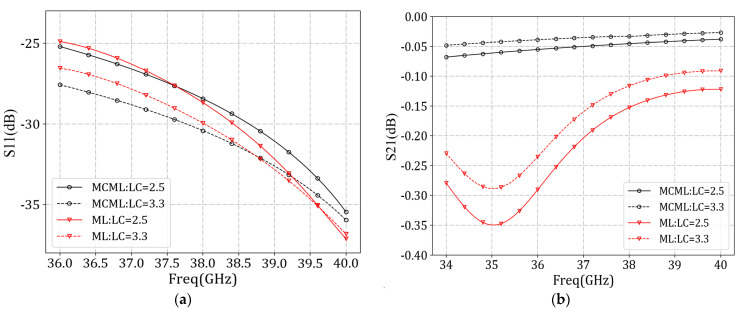
Comparison of MCML and ML performance after removing the CSRR slot. (**a**) *S*_11_ parameters; (**b**) *S*_21_ parameters.

**Figure 4 sensors-24-03467-f004:**
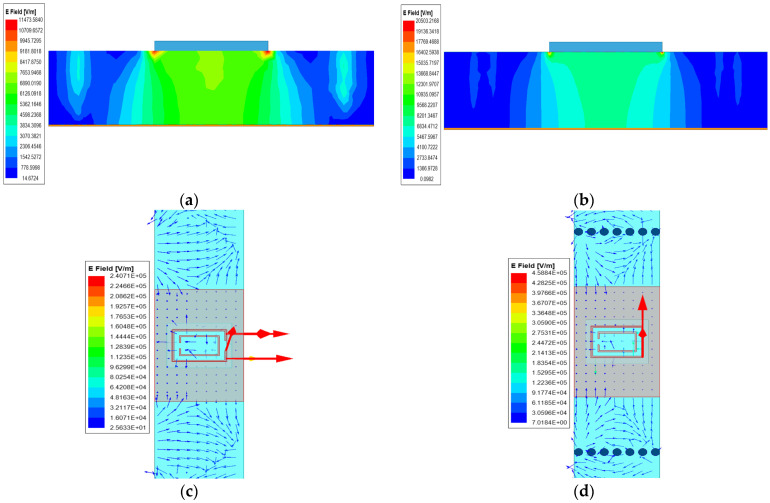
Comparison of electromagnetic wave transmission between MCML and ML structures. (**a**) ML electric field energy map; (**b**) MCML electric field energy map; (**c**) ML electric field vector map; (**d**) MCML electric field vector map.

**Figure 5 sensors-24-03467-f005:**
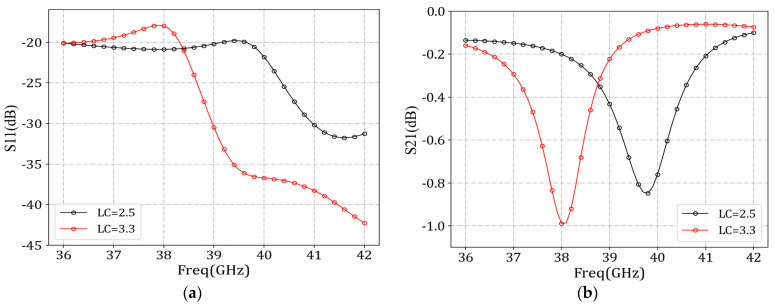
MCML cell performance simulation. (**a**) *S*_11_ parameters; (**b**) *S*_21_ parameters.

**Figure 6 sensors-24-03467-f006:**
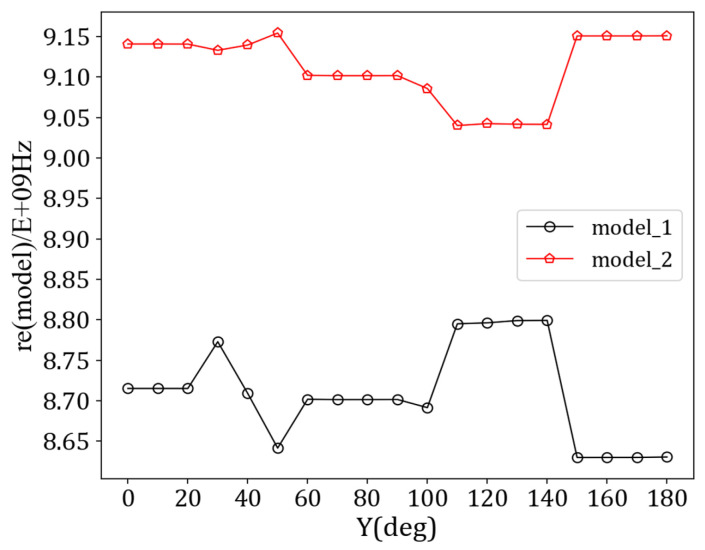
Dispersion curve of antenna unit.

**Figure 7 sensors-24-03467-f007:**
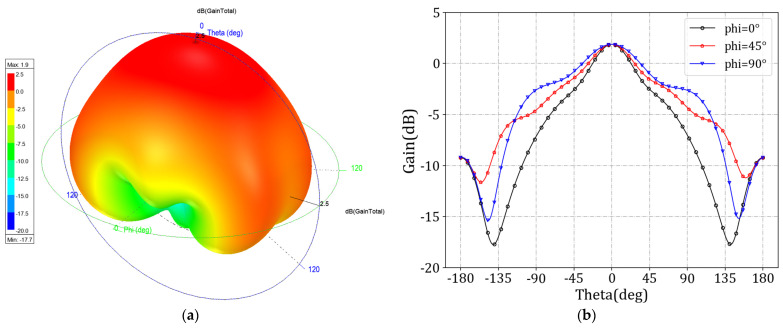
Radiation diagrams of MCML cell. (**a**) 3D plot; (**b**) 2D plot.

**Figure 8 sensors-24-03467-f008:**
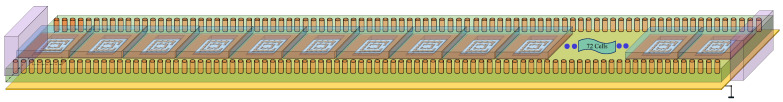
Structure of the periodic leakage antenna array.

**Figure 9 sensors-24-03467-f009:**
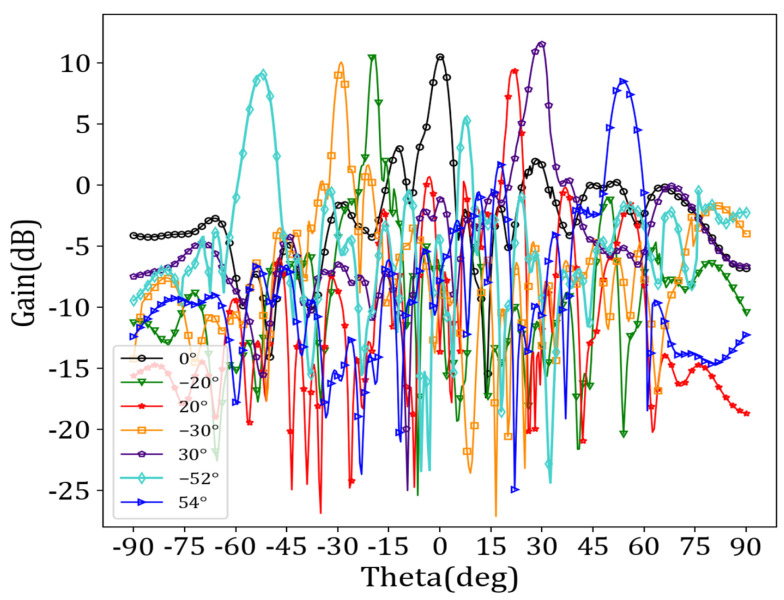
Beam gain maps corresponding to different beam directions.

**Figure 10 sensors-24-03467-f010:**
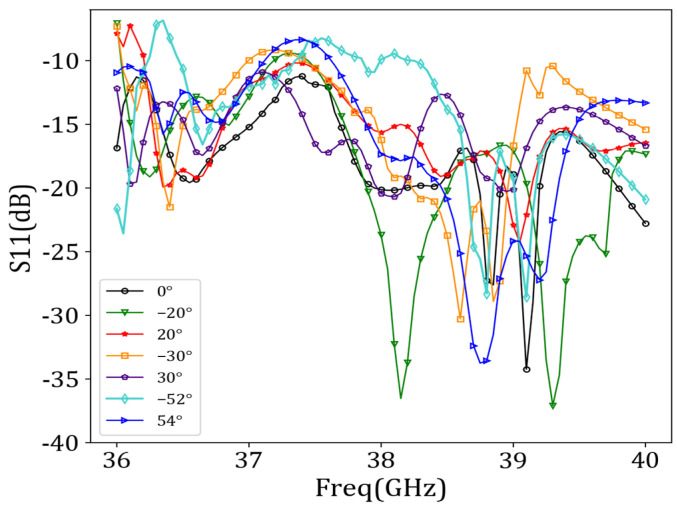
Reflection coefficients corresponding to different beam directions.

**Figure 11 sensors-24-03467-f011:**
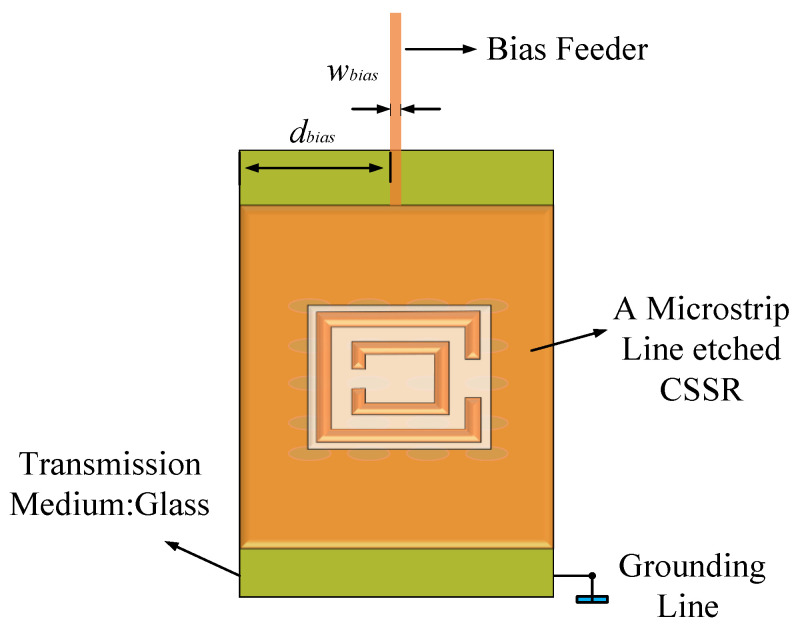
Bias feeder structure.

**Figure 12 sensors-24-03467-f012:**
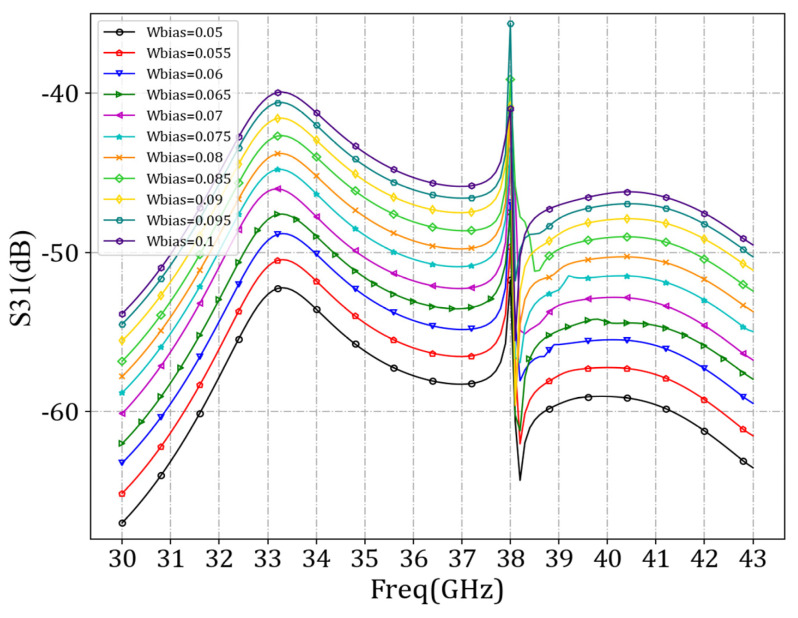
Comparison curves of S_31_ parameters under different *w_bias_* values.

**Figure 13 sensors-24-03467-f013:**
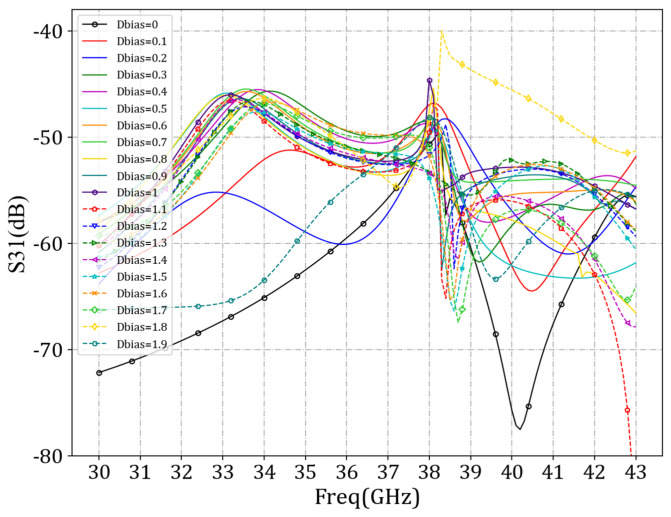
Comparison curves of *S*_31_ parameters under different *d_bias_* values.

**Figure 14 sensors-24-03467-f014:**
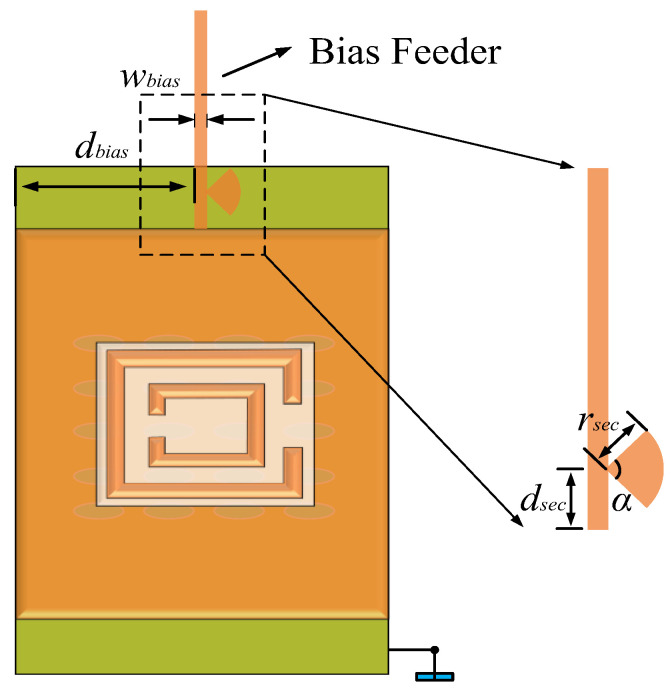
Sector-blocking bias feeder structure.

**Figure 15 sensors-24-03467-f015:**
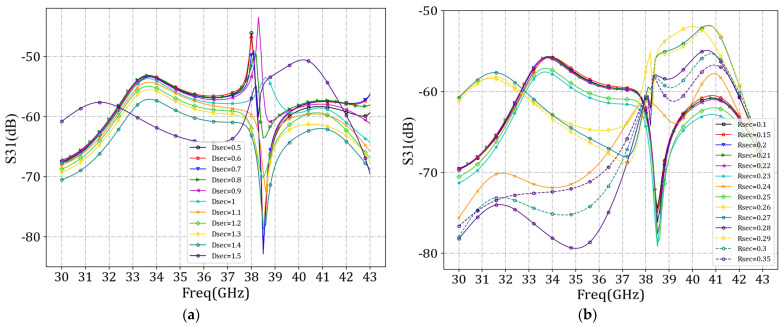
Comparison curves of *S*_31_ parameters under different *d_sec_* and *r_sec_* values. (**a**) *d_sec_*; (**b**) *r_sec_*.

**Figure 16 sensors-24-03467-f016:**
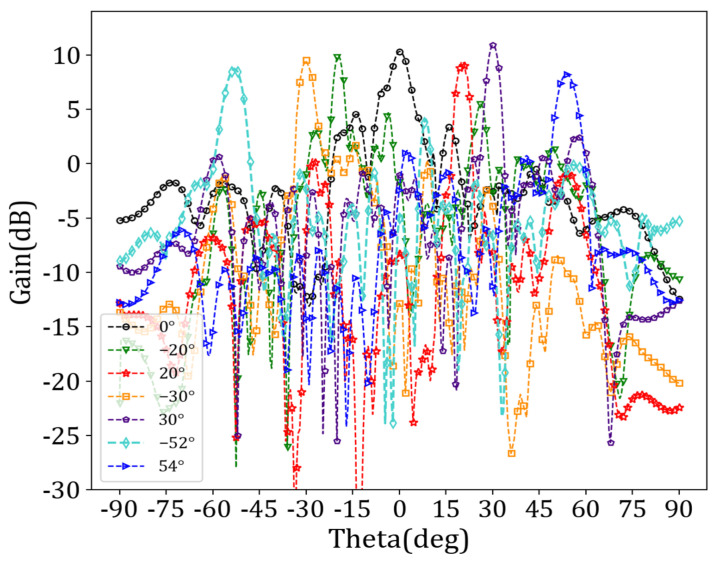
Beam gain diagrams corresponding to different beam directions after adding bias feeder.

**Figure 17 sensors-24-03467-f017:**
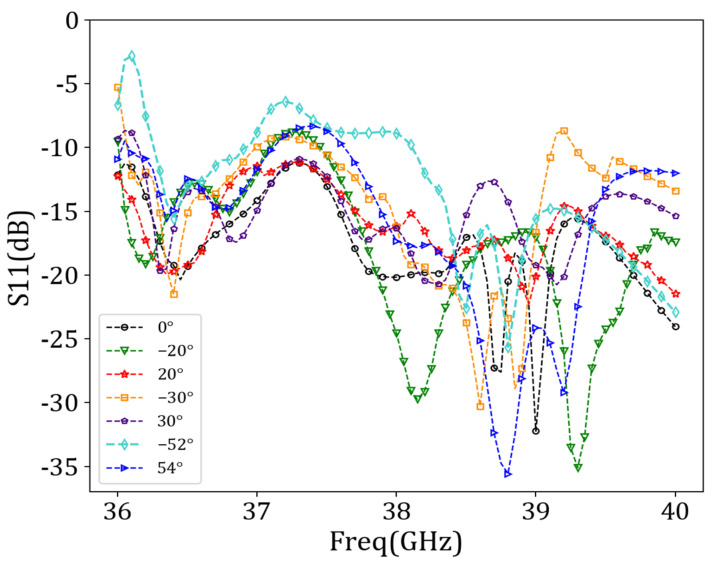
Reflection coefficients corresponding to different beam directions after adding bias feeder.

**Table 1 sensors-24-03467-t001:** Parameters of different LC models.

LC Model	$\varepsilon_{//}$	$\varepsilon_{⊥}$	τ	$\tan\delta_{//}$	$\tan\delta_{⊥}$
K15	2.57	2.3	10.5%	0.0048	0.02
E7	3.35	2.72	18.8%	0.001	0.006
TUD-566	3.16	2.4	24.1%	0.002	0.006
BL006	3.03	2.62	13.53%	0.01	0.025
MDA-05-893	2.65	2.24	15.47%	0.01	0.025
GT3-23001	3.3	2.5	24.2%	0.0038	0.0143

**Table 2 sensors-24-03467-t002:** Antenna cell parameters.

Parameters	*W*	*W* _1_	*W* _2_	*W* _3_	*W* _4_	*W* _5_
Size/mm	2	1.065	0.385	0.03	0.28	0.12
Parameters	*W* _6_	*L*	*L_b_*	*L_c_*	*L_m_*	*L_n_*
Size/mm	3	8.6	2.8	1.45	0.2	0.3
Parameters	*L_g_*	*H* _1_	*H* _2_	*H* _3_	*H* _4_	-
Size/mm	1.12	0.2	0.002	1.5	0.035	-

**Table 3 sensors-24-03467-t003:** Binary discrete amplitude-weighted sequences of 72 cells with different beam directions.

Direction	Binary Discrete Amplitude-Weighted Sequences
0°	0, 1, 0, 1, 0, 1, 0, 1, 0, 0, 1, 0, 1, 0, 1, 0, 1, 0, 0, 1, 0, 1, 0, 1, 0, 1, 0, 0, 1, 0, 1, 0, 1, 0, 1, 0, 0, 1, 0, 1, 0, 1, 0, 1, 0, 0, 1, 0, 1, 0, 1, 0, 1, 0, 0, 1, 0, 1, 0, 1, 0, 1, 0, 0, 1, 0, 1, 0, 1, 0, 1, 0
−20°	0, 1, 0, 1, 1, 0, 1, 1, 0, 1, 0, 0, 1, 0, 0, 1, 0, 1, 1, 0, 1, 1, 0, 1, 0, 0, 1, 0, 0, 1, 0, 1, 1, 0, 1, 1, 0, 1, 0, 0, 1, 0, 0, 1, 0, 1, 1, 0, 1, 1, 0, 1, 0, 0, 1, 0, 0, 1, 0, 1, 1, 0, 1, 1, 0, 1, 0, 0, 1, 0, 0, 1
20°	1, 0, 1, 0, 1, 0, 1, 0, 1, 0, 1, 0, 1, 0, 1, 0, 0, 1, 0, 1, 0, 1, 0, 1, 0, 1, 0, 1, 0, 1, 0, 1, 1, 0, 1, 0, 1, 0, 1, 0, 1, 0, 1, 0, 1, 0, 1, 0, 0, 1, 0, 1, 0, 1, 0, 1, 0, 1, 0, 1, 0, 1, 0, 1, 1, 0, 1, 0, 1, 0, 1, 0
−30°	0, 1, 0, 0, 1, 1, 0, 1, 1, 0, 1, 1, 0, 1, 1, 0, 0, 1, 0, 0, 1, 0, 0, 1, 0, 0, 1, 1, 0, 1, 1, 0, 1, 1, 0, 1, 1, 0, 0, 1, 0, 0, 1, 0, 0, 1, 0, 0, 1, 1, 0, 1, 1, 0, 1, 1, 0, 1, 1, 0, 0, 1, 0, 0, 1, 0, 0, 1, 0, 0, 1, 1
30°	1, 0, 1, 0, 1, 0, 1, 1, 0, 1, 0, 1, 0, 1, 1, 0, 1, 0, 1, 0, 1, 1, 0, 1, 0, 1, 0, 1, 1, 0, 1, 0, 1, 0, 1, 1, 0, 1, 0, 1, 0, 1, 1, 0, 1, 0, 1, 0, 1, 1, 0, 1, 0, 1, 0, 1, 1, 0, 1, 0, 1, 0, 1, 1, 0, 1, 0, 1, 0, 1, 1, 0
−52°	0, 1, 1, 0, 0, 1, 1, 0, 0, 1, 1, 1, 0, 0, 1, 1, 0, 0, 1, 1, 0, 0, 1, 1, 0, 0, 1, 1, 0, 0, 0, 1, 1, 0, 0, 1, 1, 0, 0, 1, 1, 0, 0, 1, 1, 0, 0, 1, 1, 1, 0, 0, 1, 1, 0, 0, 1, 1, 0, 0, 1, 1, 0, 0, 1, 1, 0, 0, 0, 1, 1, 0
−54°	1, 0, 1, 1, 0, 1, 1, 0, 1, 1, 0, 1, 1, 0, 1, 1, 0, 1, 1, 0, 1, 1, 0, 1, 1, 0, 1, 1, 0, 1, 0, 0, 1, 0, 0, 1, 0, 0, 1, 0, 0, 1, 0, 0, 1, 0, 0, 1, 0, 0, 1, 0, 0, 1, 0, 0, 1, 0, 0, 1, 0, 0, 1, 0, 0, 1, 0, 0, 1, 0, 0, 1

**Table 4 sensors-24-03467-t004:** Angle deviations and gains corresponding to different beam directions.

Preset Angle	−55°	−30°	−20°	0°	20°	30°	60°
Actual Angle	−52°	−29.2°	−19.5°	0.3°	21.5°	29.6°	54°
Angle Deviation	3°	0.8°	0.5°	0.3°	1.5°	0.4°	6°
Gain (dB)	9.16 dB	10.09 dB	10.67 dB	10.55 dB	9.47 dB	11.65 dB	8.52 dB

**Table 5 sensors-24-03467-t005:** The angle deviations and gains corresponding to different beam directions.

Preset Angle	−55°	−30°	−20°	0°	20°	30°	60°
Actual Angle	−51.7°	−29°	−19.4°	0.1°	21.1°	30.7°	53.8°
Angle Deviation	3.3°	1°	0.6°	0.1°	1.1°	0.7°	6.2°
Gain (dB)	8.719 dB	9.502 dB	9.931 dB	10.304 dB	9.386 dB	10.892 dB	8.216 dB

## Data Availability

Data are contained within the article.

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
