# Peer review of "MCML-BF: A Metal-Column Embedded Microstrip Line Transmission Structure with Bias Feeders for Beam-Scanning Leakage Antenna Design"

_sensors, 2024, doi:10.3390/s24113467_

Round 1

Reviewer 1 Report

Comments and Suggestions for Authors

This paper presents a design of beam-scanning LWA using LC material. However, since the CSRR unit and the beam-scanning principles of these types of LWAs have been widely investigated, the novelty of this paper is not clear. Please find the comments as follows:

1)    The CSRR unit has been widely studied in existing publications. The novelty about the structure design is not clear.

2)    It seems that the LC material is filled in separated blocks, as shown in Fig.1(a), and the LC blocks are discretely distributed rather than continuously distributed in y direction. How did you fill the LC in these blocks in the practical perfusion of LC? In addition, how can you embed metal columns into the glass? It seems difficult to be implemented in the practical manufacturing.

3)    In Section 2 the holographic theory is introduced in detail. However, in the practical design in Section 3.2, it seems that the authors were using the space-harmonic theory to realize beam scanning by controlling the period lengths. It is quite confusing which method the authors were using in the practical design. On the other hand, both of these two methods (holographic method and period-reconfigurable method) have been deeply investigated in other existing publications. Thus the novelty is not clear to me.

4)    It is not clear if there is a bias feeder for each CSRR unit or not. If yes, since the microstrip line etched layer of each unit is a continuous large metal layer without gap between adjacent units, it seems that the biasing voltage for each unit cannot be separated, and hence each individual unit cannot be controlled individually.

5)    Related with comment 4, since there is no electrical connection between two split rings of the CSRR unit and the metal layer, the biasing voltage cannot be applied to the split rings. Therefore, the molecular rotation beneath the rings will not be significant, which has impact on the 1/0 switching property.

6)    There are only five beams in the scanning pattern (Fig.15), which is too limited compared to other publications about fixed-frequency beam-scanning LWAs.

7)    It is suggested to make a practical fabrication and measurement, as a strong verification of the design.

8)    There have been so many literatures on the topic of fixed-frequency beam-scanning LWAs, which operate based on the holographic and period-reconfigurable method, using LC and diodes. However, in this paper the introduction lacks of a comprehensive overview on the recent progress of this topic.

9)    In the caption of Fig.12 “Wbias” should be “Dbias”.

Comments on the Quality of English Language

The English grammar should be checked carefully.

Reviewer 2 Report

Comments and Suggestions for Authors

This paper, proposes a novel fixed-frequency beam scanning leakage antenna based on 12 liquid crystal metamaterial (LCM) and adopts a metal-column embedded microstrip line (MCML) 13 transmission structure. There are some points need to be improved.

1. the abstract should focus on the novelty in the paper compared to similar work.

2. what are the benfits of using LC over other reconfigurable techniques such as pin diodes, graphene and plasma.

3. the introduction should be improved to include the state of art of other reconfigurable techniques used in LWA. these references should be help

a.Malhat, Hend A., Abdelkarim S. Elhenawy, Saber H. Zainud-Deen, and Noha A. Al-Shalaby. "Planar reconfigurable plasma leaky-wave antenna with electronic beam-scanning for MIMO applications." Wireless Personal Communications 128, no. 1 (2023): 1-18.

b. Esquius-Morote, Marc, Juan Sebastian Gómez-Dı, and Julien Perruisseau-Carrier. "Sinusoidally modulated graphene leaky-wave antenna for electronic beamscanning at THz." IEEE Transactions on Terahertz Science and Technology 4, no. 1 (2014): 116-122.

c. Cheng, Y., Wu, L.S., Tang, M., Zhang, Y.P. and Mao, J.F., 2017. A sinusoidally-modulated leaky-wave antenna with gapped graphene ribbons. IEEE Antennas and Wireless Propagation Letters, 16, pp.3000-3004.

d. Soleimani, Hadi, and Homayoon Oraizi. "A novel 2D leaky wave antenna based on complementary graphene patch cell." Journal of Physics D: Applied Physics 53, no. 25 (2020): 255301.

e.Al-Shalaby, Noha A., Abdelkarim S. Elhenawy, Saber H. Zainud-Deen, and Hend A. Malhat. "Electronic beam-scanning strip-coded graphene leaky-wave antenna using single structure." Plasmonics 16 (2021): 1427-1438.

f. Cao, Xiaowei, Changjiang Deng, and Kamal Sarabandi. "Fixed-frequency beam steering leaky-wave antenna with integrated 2-bit phase shifters." IEEE Transactions on Antennas and Propagation 70, no. 11 (2022): 11246-11251.

g. Du, H., Wang, J., Li, Z., Zheng, W. and Zhang, L., 2023. A Broadband Fixed-Beam Leaky-Wave Antenna with Switchable Beam Direction. IEEE Antennas and Wireless Propagation Letters.

h.  Zheng, Wei, Junhong Wang, Huiying Zhao, Zheng Li, Yunjie Geng, Yujian Li, Meie Chen, and Zhan Zhang. "A leaky-wave antenna with capability of fixed frequency beamforming scanning." IEEE Transactions on Antennas and Propagation (2023).

4. The dispersion diagram of the unit-cell should be added.

5.the authors stated that "Beam scanning capability is realized based on the holographic antenna principle" but the design curve of Zsurf of the LCM is not shown.

6. what is the variation parameter from a unit cell to another in the same period (Zsurf and corresponding voltage)

7. what are the DC voltage values range used in cell tuning

Comments on the Quality of English Language

none

Round 2

Reviewer 1 Report

Comments and Suggestions for Authors

The author did not response to some of my comments actively and positively, especially regarding comments 1, 2, and 7. Nevertheless, I found that they have already made modifications to address other questions. Perhaps they have already tried their best. Thus acceptance is suggested.

Comments on the Quality of English Language

N/A